# All-Organic PTFE Coated PVDF Composite Film Exhibiting Low Conduction Loss and High Breakdown Strength for Energy Storage Applications

**DOI:** 10.3390/polym15051305

**Published:** 2023-03-05

**Authors:** Xiang-Shuo Meng, Yujiu Zhou, Jianfeng Li, Hu Ye, Fujia Chen, Yuetao Zhao, Qifeng Pan, Jianhua Xu

**Affiliations:** 1Ocean College, Jiangsu University of Science and Technology, Zhenjiang 212100, China; 2Department of Film Capacitors, China Zhenhua (Group) Xinyun Electronic Components and Development Company Limited, Guiyang 550018, China; 3School of Optoelectronic Science and Engineering, University of Electronic Science and Technology of China, Chengdu 610054, China

**Keywords:** dielectric film, energy density, low loss, conductance mechanism

## Abstract

Plastic film capacitors are widely used in pulse and energy storage applications because of their high breakdown strength, high power density, long lifetime, and excellent self-healing properties. Nowadays, the energy storage density of commercial biaxially oriented polypropylene (BOPP) is limited by its low dielectric constant (~2.2). Poly(vinylidene fluoride) (PVDF) exhibits a relatively high dielectric constant and breakdown strength, making it a candidate material for electrostatic capacitors. However, PVDF presents significant losses, generating a lot of waste heat. In this paper, under the guidance of the leakage mechanism, a high-insulation polytetrafluoroethylene (PTFE) coating is sprayed on the surface of a PVDF film. The potential barrier at the electrode–dielectric interface is raised by simply spraying PTFE and reducing the leakage current, and then the energy storage density is increased. After introducing the PTFE insulation coating, the high-field leakage current in the PVDF film shows an order of magnitude reduction. Moreover, the composite film presents a 30.8% improvement in breakdown strength, and a 70% enhancement in energy storage density is simultaneously achieved. The all-organic structure design provides a new idea for the application of PVDF in electrostatic capacitors.

## 1. Introduction

With the development of electrical devices, the requirements for energy storage components become more urgent. As a passive component, capacitors are wildly used in new energy vehicles, as pulse power, and as defibrillators [1,2,3,4]. Compared to ceramic capacitors and supercapacitors, plastic film capacitors possess superior stability, require simple preparation, have excellent mechanical performance, high breakdown strength, high power density (10^8^ W/kg), and high charge–discharge efficiency [5,6,7].

The most important part of electrostatic capacitors is the dielectric material, which directly determines the electric properties. Biaxially oriented polypropylene (BOPP) is the most commonly used commercial dielectric material because of its high breakdown strength, good charge–discharge efficiency, low dielectric loss, and long service life [8,9,10,11,12]. However, the dielectric constant of BOPP is only 2.2 at 1 kHz, which severely limits its energy storage density. Low energy density leads to a heavy volume of the capacitor and adds additional manufacturing costs. For example, polymer dielectric capacitors always occupy 50% of volume and a 40% mass fraction in a pulsed-power system [13,14,15,16]. The energy storage density of a dielectric material can be calculated using the following equation [6,17]:(1)U=∫EdD

As for the linear dielectric material, the energy density can be simplified as follows:(2)$$U=\frac{1}{2}\varepsilon_{0}\varepsilon_{r}E^{2}$$
where *E* is the applied electric field and *D* is the electric displacement, *ε_o_* is the vacuum dielectric constant, and *ε_r_* is the relative dielectric constant. Poly(vinylidene fluoride) (PVDF) is a kind of ferroelectric polymer with high spontaneous dipole (pointing from the fluorine atom to the hydrogen atom) [18,19], exhibiting a high dielectric constant (~8–12), high breakdown strength, and excellent energy storage density. However, because PVDF has a spontaneous dipole, there is a significant hysteresis effect while the applied electric field is above its coercivity field (β-phase is 70 kV/mm). Therefore, PVDF is difficult to be used in AC electrostatic capacitors [20]. In general, the energy storage efficiency of PVDF is less than 60%, which means that extreme dielectric losses are converted into waste heat and even lead to thermal runaway, affecting the lifetime of the capacitor [21].

There has been significant progress in research on energy storage for dielectric materials, but the energy density is often enhanced at the expense of charge–discharge efficiency or degradation loss. In order to increase the energy density, the usual method used is to modify the dielectric material with high dielectric constant fillers. Bouharras [22] increased the dielectric constant of PVDF by 50% through preparing a core–shell structure of BaTiO_3_ and grafting it onto PVDF, but creating a certain increase in loss as well. Xu [23] filled the PVDF with Ni, resulting in a significant simultaneous enhancement in the dielectric constant and the leakage loss by nearly an order of magnitude. The introduction of high dielectric constant fillers is often accompanied by an extremely high leakage current. Materials with two widely different dielectric constants are also prone to large electric field distortions, leading to the premature breakdown of the dielectric film. High dielectric constant fillers are usually ceramics or metals, and the organic–inorganic system suffers a sharp deterioration of mechanical properties. For capacitor applications, reducing losses in polymer dielectric films is as important as increasing the dielectric constant. The heat generated by dielectric losses can weaken insulation, which will lead to early failure, and even cause polymer films to catch fire [24].

Loss is one of the key parameters to determine whether a dielectric film can be used in electrostatic capacitors. Chiu [25] proposed a classification of leakage conduction loss, which includes two major mechanisms of electrode-limited conductance and bulk-limited conductance. Xu [26] controlled the morphology (grain structure, grain size, surface roughness) of BOPP films by different annealing temperatures and demonstrated that the leakage mechanism of polypropylene films belongs to jump conductance in bulk-limited conductance with an intrinsic jump distance of about 0.1–1.2 nm. The dependence of the jump distance on microscopic parameters was further demonstrated. Yuan [27] introduced the antioxidant group, hindered phenol (HP), into PP through an easily controlled chemical synthesis to form a cross-linking network, which inhibits thermal decomposition and charge transport in PP–HP copolymers and reduces the conduction loss of PP films. Zhu [28] investigated the mechanism of electron conduction in biaxially oriented PVDF at a high field. Thermally stimulated depolarization current (TSDC) spectroscopy and leakage current studies demonstrate that space charge injection from the metal electrode is the primary origin of electron conduction, when PVDF is polarized at 75 °C under an electric field higher than 20 kV/mm. Additionally, based on the dielectric temperature spectra, it is shown that the relatively high dielectric constants of PVDF are caused by amorphous dipole polarization. This demonstrates that the loss in PVDF is related to the electrode-dielectric interface and crystallinity. Huang [29] prepared a PVDF composite dielectric film using the solution casting method, which includes PVDF as the outer layer and a nano-sized intermediate layer of boron nitride nanosheets (BNNSs) arranged along the in-plane direction. The skinny two-dimensional nano-sized middle layer of BNNSs blocks the charge transport path and effectively suppresses the leakage current of the composite film.

Zhou [30] designed a poly(methyl methacrylate) (PMMA) film with a multilayer structure, which significantly improved the charge–discharge efficiency and reduced the conduction loss. Chen [31] found that the doping of wide-bandgap boron nitride nanosheets in fluorine-containing copolymers extends the electron conduction path and blocks the growth of electric dendrites, which successfully suppresses the conduction loss. However, compatibility is an issue that has to be faced directly, regardless of the multilayer structure or the filler.

In this paper, we inhibit electron injection and reduce leakage current by preparing a high insulation polytetrafluoroethylene (PTFE) coating at the electrode–dielectric interface. The effect of the annealing temperature on the PVDF composite dielectric film was also investigated. Compared with adding inorganic fillers within the polymer, the direct spraying of organic PTFE has the advantage of being simple and easy to operate, while the all-organic system maintains excellent mechanical performance. The dielectric constant of the high insulation PTFE coating is much smaller than that of the PVDF film, so the electric field is mainly concentrated in the PTFE layer. As a result, the breakdown strength of the composite film after treatment at 80 °C reaches a maximum of 411 kV/mm, which is 30.8% higher than that of the pristine PVDF film. Moreover, the high insulation PTFE coating inhibits electron injection from the electrode and suppresses the conduction loss. The leakage current is reduced by one order of magnitude at a 300 kV/mm field strength for the film treated at 60 °C. Owing to the high breakdown strength and low leakage current of the composite, the charge–discharge efficiency remains above 79%, and the energy storage density is increased by up to 70% simultaneously. This study offers an easy route for preparing a low loss PVDF-based composite.

## 2. Materials and Methods

### 2.1. Materials and Equipment

The PVDF film was purchased from Polyk, and the PTFE suspension was provided by Rocol. The thickness of the PVDF and its composite film was tested using the Fischer coating thickness gauge. A 12 mm × 12 mm square aluminum electrode and a 2-mm diameter round aluminum electrode were vaporized on the surface of the dielectric film for the characterization of the electrical properties. The thickness of the metal layer is about 100 nm. The permittivity was acquired at room temperature via an Agilent impedance analyzer type 4294A with a frequency range from 40 Hz to 5 MHz. The leakage currents and the polarization–electric field (P–E) loops were measured using Radiant multiferroic technology with a Trek high voltage amplifier (609 E). For the leakage current measurement, the test time and the soak time were set as 20,000 ms and 200 ms, respectively. The period of the triangular drive profile was set as 100 ms in the P-E loops test. The breakdown strength was obtained using a high voltage insulation analyzer (TH 9201) with a 500 V/s voltage rise rate, and the upper limit of the test current was set as 5 mA. Each breakdown sample contains 16 breakdown strength points to ensure accuracy.

### 2.2. Preparation of PTFE-Coated Composite Films

The PTFE spray was shaken before application to make the suspension uniform, and an A4 paper was rolled into a tube and fixed at a position 5 cm above the PVDF film. The PTFE suspension was sprayed from the top of the A4 paper barrel to deposit PTFE on one side of the PVDF film, as shown in Figure 1, and the PVDF film deposited with the PTFE suspension was put into an oven at 60 °C drying for 0.5 h. Then the other side was sprayed via the same method. In this paper, the PTFE-coated PVDF composite film is defined as a PTFE–c–PVDF composite film. Appendix A shows the FTIR spectra of PVDF and PTFE–c–PVDF. It proved that the PTFE coating was effectively prepared on the surface of the PVDF film. For simplicity, the drying temperature (60 °C, 80 °C, and 100 °C) was changed to prepare various samples, named as PT60, PT80, and PT100, respectively. For comparison, the same heat treatment for the pure PVDF was carried out simultaneously, and was named P60, P80, and P100.

## 3. Results and Discussion

### 3.1. Loss of PVDF and PTFE-c-PVDF 

As shown in Figure 2, the leakage current of the pure PVDF film stays almost steady from 70 kV/mm to 170 kV/mm, and exponentially increases above 170 kV/mm. This involves the establishment of an electric field inside the PVDF polymer. When an electric field is applied to the PVDF dielectric film, three kinds of built-in electric fields appear inside [28], i.e., the polymer internal heterogeneous charge polarization, internal dipole polarization, and electrode injection charge. The polymer internal dipole polarization and the heterodyne charge form a built-in electric field which shows the opposite direction of the applied electric field. The injected charge will create a built-in electric field in the same direction as the applied electric field. This is the main reason that the leakage current keeps the balance under a relatively low electric field. With the increase in the applied electric field, more and more injected electrons from the electrode, de-trapping electrons, and tunneling electrons turn up, leading to an enhancement in the leakage current. The establishment of a built-in electric field inhibits the growth of leakage current, and the stabilization of leakage current is the result of the balance between these two factors. At the limit where the applied electric field increases to equilibrium, the leakage current begins to rise exponentially.

Compared with the pure PVDF dielectric film, the PTFE–c–PVDF composite film possesses a slightly lower leakage current under an electric field of less than 200 kV/mm. At low electric fields, the primary conductance mechanism is Ohmic conductance, which is determined by the carrier concentration and carrier mobility inside the dielectric. The PTFE insulation coating mainly affects the electrode-limited conductance and has little influence on the carrier concentration and carrier mobility inside the dielectric. As a result, the leakage current is only slightly reduced in the 200 kV/mm electric field. In high electric fields, the conduction loss of PVDF is mainly attributed to the electrode-limited conductance, which is primarily generated via electrons injected from the electrode into the dielectrics. The PTFE coating with an extremely high insulation performance can effectively increase the Schottky barrier between the electrode and polymer, inhibiting electron injection from the electrode [31]. As a result, the leakage current of the PTFE–c–PVDF composite film under a high field significantly decreases, especially when the electric field is higher than 200 kV/mm, as shown in Figure 2a. The leakage current of the PVDF dielectric films treated at different temperatures shows distinct trends. The leakage current of PVDF films treated at 60 °C and 80 °C decreases, while the leakage current of PVDF dielectric films treated at 100 °C deteriorates sharply under an electric field above 170 kV/mm and breaks down under a 230 kV/mm electric field. The high-temperature treatment of P100 leads to the internal crystalline region relaxation process of PVDF, which makes the leakage current increase in high fields [32]. The heat generated by the loss of the dipole will degrade the film quality and cause premature breakdown of the film. As for other composite films, a proper temperature could promote volatilization of the solvent in the PTFE suspension, and makes the PTFE more uniformly distributed on the surface of PVDF.

As shown in Figure 2b, the charge–discharge efficiency of PVDF and its composite films was tested. In the PTFE–c–PVDF composite film, the charge–discharge efficiency is higher than that of pristine PVDF films. This indicates that the high insulation PTFE coating effectively suppresses the electron injection from the electrode, reducing the conduction loss under a high electric field. The charge–discharge efficiency of PT100 decreases sharply because the dipole inside the PVDF relaxes at 100 °C under the influence of high temperature. The relaxed dipole generates more relaxation losses under the action of the applied electric field, reducing the charging–discharging efficiency.

The polarization–electric field (P–E) loops of PVDF and its composite films were shown in Figure 3. The hysteresis loop of pure PVDF becomes fatter after 100 °C treatment, indicating that the energy loss increases. The PVDF films are prepared via melt extrusion biaxial stretching. During the biaxial stretching process, the stress makes the dipoles align along the stretching direction. The ordered dipoles reduce the relaxation period when the electric field is applied and maintain the excellent performance of the pristine PVDF film. In PT100, the neatly aligned dipoles in the PVDF film relax to disorder after the 100 °C treatment. Then the relaxation loss increases when an applied electric field is applied, generating a large amount of waste heat and reducing the performance of the PVDF film, resulting in a fat hysteresis line. Figure 4 shows the polarization diagram of PVDF and its composite films. For the pure PVDF film, there is a slight increase in maximum polarization after heat treatment at 60 °C and 80 °C, respectively. The maximum polarization of the PTFE–c–PVDF composite film is similar to that of the pure PVDF. It is well known that a higher efficiency represents a lower loss during the charging–discharging process. In ferroelectric materials, the losses are mainly included in ferroelectric losses and conduction losses. The ferroelectric losses are primarily caused by the ferroelectric domains, and the conduction losses are mostly triggered by the leakage current generated by the carrier motion inside the material. So, the ferroelectric losses originate from polarization hysteresis, and conduction losses are mainly controlled by the conductivity of the material [32,33,34]. To better understand the source of efficiency improvement in the PTFE–c–PVDF composite film, the conduction loss and ferroelectric loss in the hysteresis loop are studied separately. It is assumed that the displacement in the absence of an applied electric field (DE = 0) is primarily from the leakage current. The effective conductivity during the charging–discharging process can be calculated using the following equation.
(3)$$P_{E=0}=\frac{1}{2}\sigma_{eff}E_{\max}T$$
where *P****_E_***_=0_ is the remanent polarization, *E*_max_ is the maximum applied electric field, and *T* is the electric field period. The effective conductivity ***σ**_eff_*** can be calculated by using the tested parameters. Based on the equation, ferroelectric loss and conduction loss can be separated [35].

From Figure 5, both the ferroelectric losses and conduction losses decrease in the PTFE–c–PVDF composite film. On the one hand, the PVDF with a high dielectric constant will bear a lower electric field in this multilayer structure based on the series model, which caused lower ferroelectric losses under the same applied electric field. On the other hand, the PTFE barrier could block the injection of carriers from the electrode, which decreases the conduction losses.

### 3.2. Breakdown Strength of PVDF and PTFE–c–PVDF

Breakdown strength is an important parameter in dielectric capacitors. According to the energy storage density equation, it is known that the breakdown strength plays an important role in the energy storage density of capacitors. The breakdown strength of the composite film is investigated using a two-parameter Weibull statistical distribution with the following equation:(4)$$P(E)=1-\exp[-{\left(E/E_{B}\right)}^{\beta }]$$
where *P*(*E*) is the cumulative failure probability, *E* is the measured breakdown strength, *E_B_* represents the Weibull breakdown strength, and *β* is the shape parameter, which can reflect the distribution of the measured electric field directly. [36,37]. Figure 6 shows the breakdown performance of PVDF and its composite film. The pure PVDF shows a breakdown strength of about 311 kV/mm, and a slight variation can be obtained after the PVDF is treated with different temperatures. The PTFE–c–PVDF composite film presents an increased breakdown strength. For instance, the breakdown strength of PT80 shows an excellent breakdown strength of 407 kV/mm, which is 30.8% higher than that of pristine PVDF. This phenomenon is mainly attributed to two reasons: (1) PTFE contains more Fluorine atoms with high electron affinity energy, which makes the Schottky barrier between electrodes and dielectrics increase. The high barrier could inhibit electron injection from the electrode and hinder the growth of electric dendrites; and (2) the PTFE–c–PVDF composite film is equivalent to a sandwich structure, in which the electric field of the film is related to their dielectric constant. It can be given using the equivalent series equation:(5)$$\varepsilon_{1}E_{1}=\varepsilon_{2}E_{2}$$

PTFE has a lower dielectric constant compared to PVDF, so PTFE will tolerate a higher electric field and decrease the electric field on PVDF.

Figure 7 shows the electric field distribution simulation result via the COMSOL Multiphysics simulation. Figure 7b presents the electric field data plotted for the cross-section of the composite dielectric film along the normal direction. It can be seen that the electric field in PTFE is six times that in PVDF, which is well agreed with the equivalent series model.

### 3.3. Energy Storage Performance of PVDF and PTFE–c–PVDF

According to the energy storage density Equation (1) and electric displacement formula,
(6)$$D=\varepsilon_{0}E+(\varepsilon_{r}-1)E$$
the polarization is mainly related to the dielectric constant. Figure 8 and Appendix A show the dielectric constant of PVDF and its composite film. The permittivity is plotted at the frequency of 1 kHz. It is found that the dielectric constant of the PTFE–c–PVDF composite film is slightly higher than that of the pure PVDF dielectric film. This is because there is an interface between the PTFE coating layer and PVDF substrate material, which would generate the interface polarization. The dielectric relaxation found in Figure 8 is because the dipoles in PVDF find it hard to follow the high-frequency transformation of the electric field. Hence, the contribution of dipoles to the dielectric constant decreases with the increase in frequency.

Figure 9 and Appendix A corroborate the accumulation of space charge at the interface via the COMSOL Multiphysics simulation, which could explain the anomalous phenomenon that the dielectric constant increases after the introduction of the low dielectric constant layer PTFE. The accumulated charge at the PTFE–c–PVDF interface is consistent with a space charge polarization mechanism. The increase in the applied electric field reduces the Schottky barrier at the PTFE–PVDF interface, making the electron injection rate greater than the internal carrier recombination rate of PVDF. There are extra electrons after the injected electrons fill the traps, accumulating at the interface to form a space charge. At this point, the space charge forms a built-in electric field opposite to the applied electric field, which can alleviate the electrode electron injection to a certain extent and suppress the generation of a leakage current. Appendix A shows that the electric field of the PTFE–c–PVDF composite film is enhanced more significantly, and the breakdown field of the composite after treatment at 80 °C is enhanced by more than 30%.

Figure 10 shows the energy storage density of PVDF and its composite. It can be obtained that the energy storage density has significantly improved in the PTFE–c–PVDF composite film with a proper annealing temperature. For instance, the energy storage density of PT80 is about 1.68 times that of pristine PVDF. The increase in temperature is beneficial to the dispersion of PTFE on the surface of PVDF and the bonding force between PTFE and PVDF. The dipoles in the dielectric film prepared via biaxial stretching are arranged in a certain direction. When the temperature reaches 100 °C, the arrangement is relaxed, leading to a decrease in the breakdown strength. Thus, the energy storage density is lower than 80 °C.

## 4. Conclusions

The PTFE–c–PVDF composite film was prepared using a simple spraying method. After a certain temperature treatment, the dielectric properties of the composite films are significantly enhanced. The high insulation PTFE coating can increase the height of the Schottky barrier between the electrodes and dielectrics, which reduces the injection of electrons from the electrodes and significantly suppresses the leakage current under high fields. The interface increases the dielectric constant and breakdown strength simultaneously. As a result, in the treated sample PT80, an energy density of 8.39 J/cc can be obtained, which is 70% higher than that of uncoated PVDF. In addition, after treatment, the breakdown strength enhanced by 30.8%, and the leakage current decreased by an order of magnitude while the charge–discharge efficiency remained above 79%. This simple coating and temperature treatment can be easily applied in the actual capacitor production process.

## Figures and Tables

**Figure 1 polymers-15-01305-f001:**
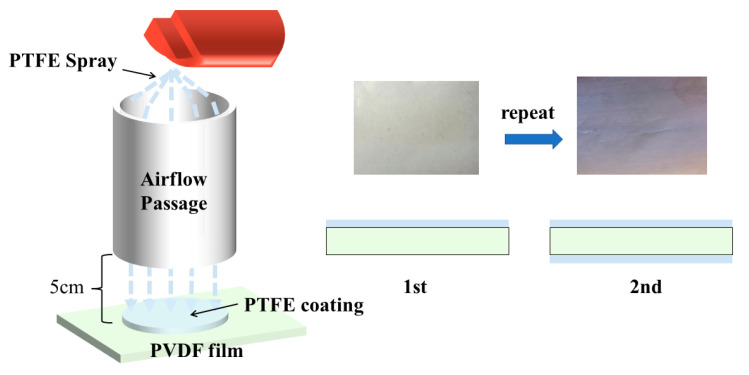
The schematic preparation process of PTFE–c–PVDF films.

**Figure 2 polymers-15-01305-f002:**
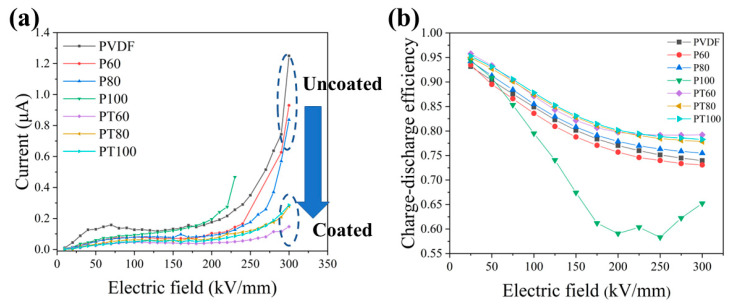
(**a**) Leakage current and (**b**) charge–discharge efficiency of PVDF and PTFE-c-PVDF.

**Figure 3 polymers-15-01305-f003:**
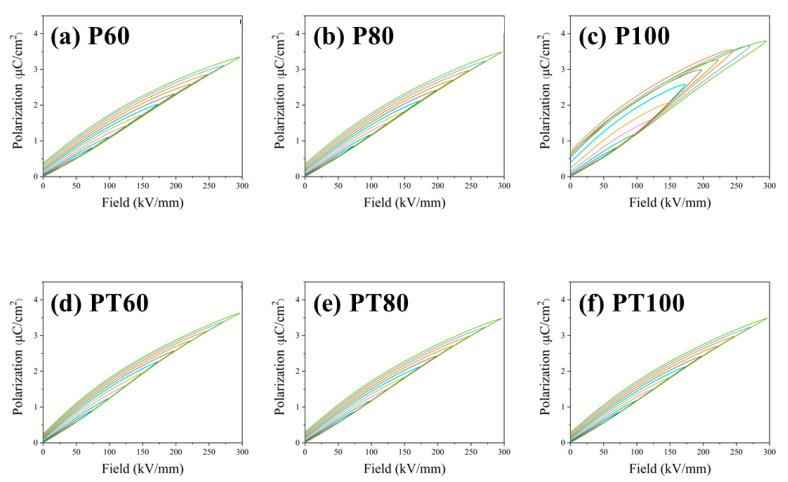
P–E loop of PVDF and its composite, (**a**–**c**) are the hysteresis loops of pure PVDF at 60, 80 °C, 100 °C, and (**d**–**f**) are the hysteresis loops of PTFE–c–PVDF at 60 °C, 80 °C, 100 °C.

**Figure 4 polymers-15-01305-f004:**
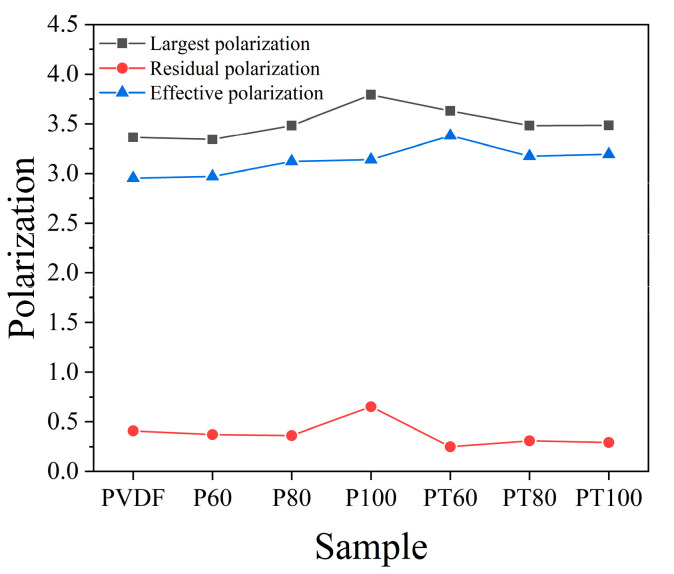
Polarization of PVDF and PTFE–c–PVDF.

**Figure 5 polymers-15-01305-f005:**
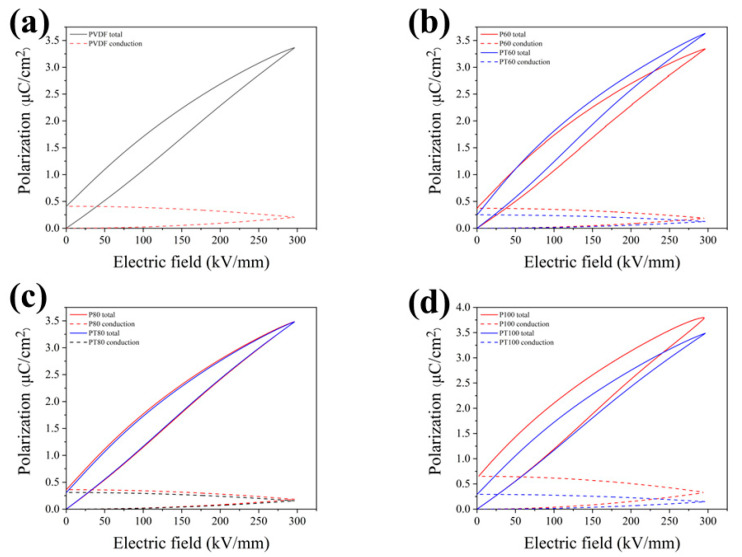
Deconvolution of charge transport contribution from P–E loops, i.e., P-E loops and conduction loss of (**a**) PVDF, (**b**) P60 and PT60, (**c**) P80 and PT80, (**d**) P100 and PT100.

**Figure 6 polymers-15-01305-f006:**
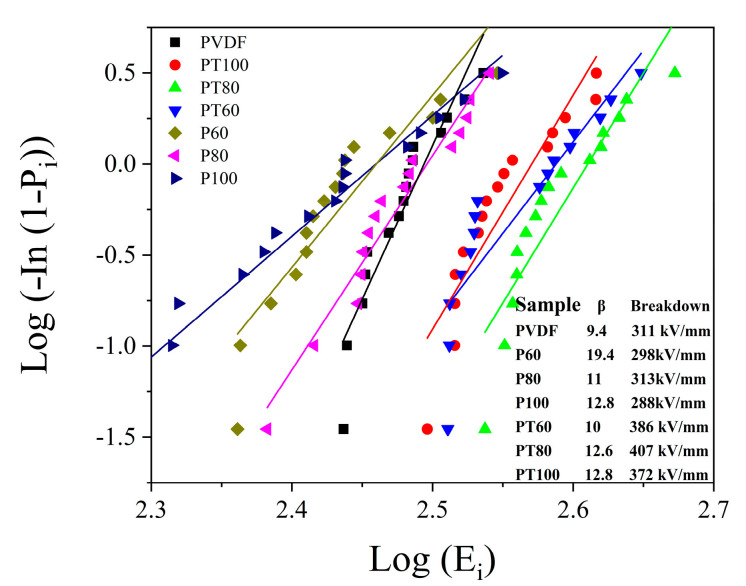
Breakdown Weibull diagram and breakdown scatter diagram.

**Figure 7 polymers-15-01305-f007:**
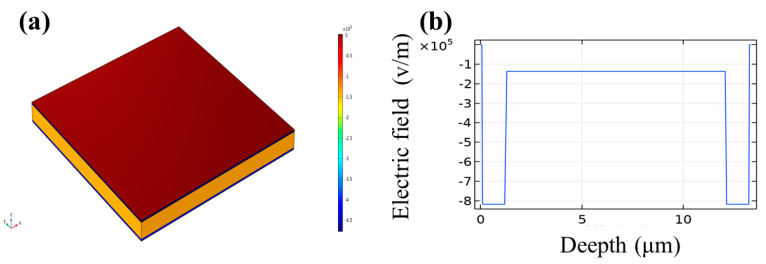
(**a**) The electric field distribution simulation result for the PTFE–c–PVDF composite film. (**b**) The electric field distribution simulation result of the cross-section along the normal direction.

**Figure 8 polymers-15-01305-f008:**
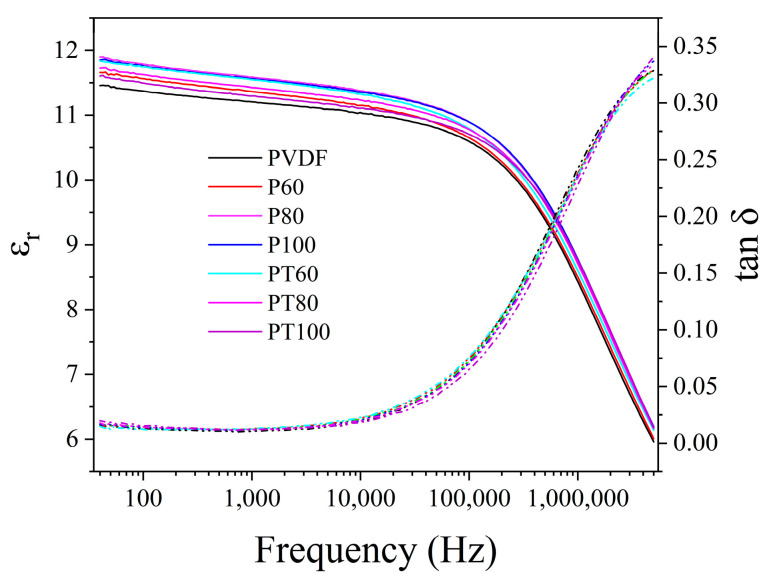
Dielectric constant and loss factor of PVDF and its composite.

**Figure 9 polymers-15-01305-f009:**
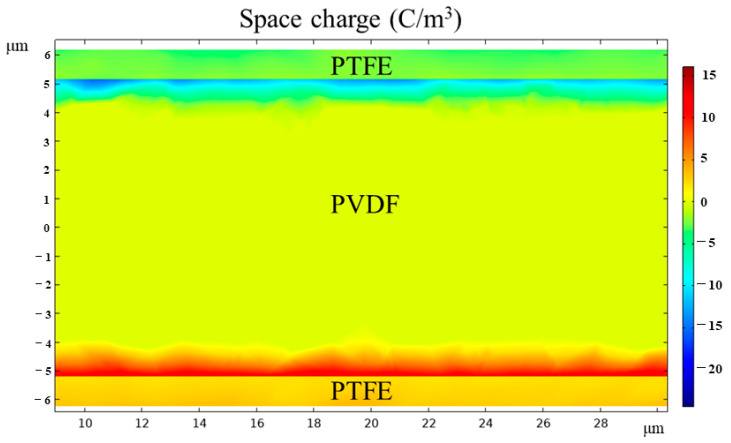
Space charge simulation of PTFE–c–PVDF composite film.

**Figure 10 polymers-15-01305-f010:**
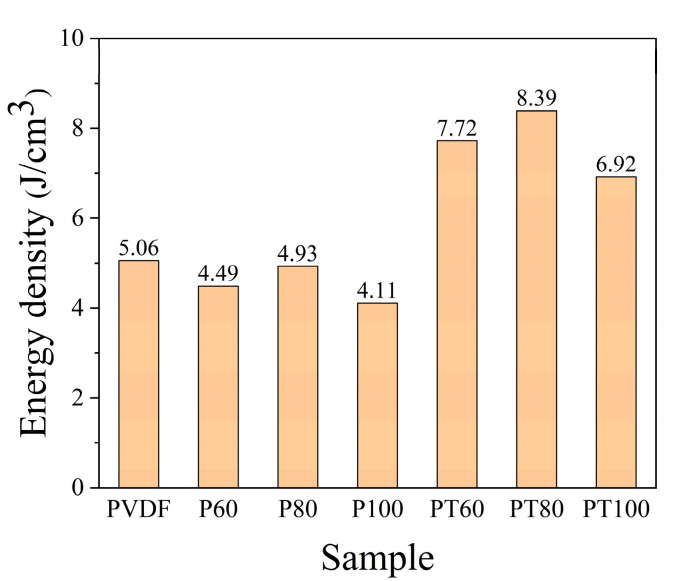
Energy storage density of PVDF and PTFE–c–PVDF.

## Data Availability

Data will be made available upon request. The raw/processed data required to reproduce these findings cannot be shared at this time as the data also form part of an ongoing study.

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
