# Peer review of "All-Organic PTFE Coated PVDF Composite Film Exhibiting Low Conduction Loss and High Breakdown Strength for Energy Storage Applications"

_polymers, 2023, doi:10.3390/polym15051305_

Round 1

Reviewer 1 Report

1) Revise and give SEM micrograph to identify defects or micro streamers ways during breakdown. In such way it can be appreciate the value of your work

2) In introduction is not convincing if your work has original contribution. Please give in references that relevant references

Reviewer 2 Report

In this work, the authors reported all-organic PTFE-coated PVDF composite film exhibiting low 2 conduction loss and high breakdown strength for energy storage 3 applications. The overall results were interesting. The Introduction was well-organized and reviewed. Some points must be revised clearly before publication, as follows:

1. For the dielectric measurement, more details must be provided such as the preparation method of an electrode used, Vrms, Frequency, or temperature range for the measurement.    

2. Figure caption of Fig. 3 must be revised, a-f must be explained in the caption.

3. Fig. 6, for the fitting data, R^2 should be presented. 

4. Fig. 8, dielectric relaxation was observed. The mechanism inside the samples must be discussed. 

5. Finally, Fig. 10, the energy density of the PT80 was the highest, please discuss and explain.

Reviewer 3 Report

The topic of the paper is interesting.  In general, the main concept of the work is interesting but Authors should improved some elements – all suggestions are given below in more detail:

1)             Abstract of the paper should be supplemented with the novelty of the topic.

2)             References should contain more open acces articles, e.g. from journals such as Marerials or Molecules, but not only from MDPI.

3)             Conclusions should be more quantified.

4)             Authors should make some editorial corrections.

Round 2

Reviewer 1 Report

Well done

Manuscript well refurbished.